# Multi-Target Neural Differentiation (MTND) Therapeutic Cocktail to Suppress Brain Tumor

**DOI:** 10.3390/ijms241512329

**Published:** 2023-08-02

**Authors:** Xiaoping Hu, Jingdun Xie, Yilin Yang, Ziyi Qiu, Weicheng Lu, Xudong Lin, Bingzhe Xu

**Affiliations:** 1School of Biomedical Engineering, Sun Yat-sen University, Guangzhou 510275, China; huxp3@mail2.sysu.edu.cn (X.H.); yangylin7@mail2.sysu.edu.cn (Y.Y.); 2Department of Anesthesiology, Sun Yat-sen University Cancer Center, State Key Laboratory of Oncology in Southern China, Collaborative Innovation for Cancer Medicine, Guangzhou 510060, China; xiejd6@mail.sysu.edu.cn (J.X.); luwc@sysucc.org.cn (W.L.); 3School of Biomedical Engineering (Shenzhen), Shenzhen Campus of Sun Yat-sen University, Shenzhen 518107, China; qiuzy8@mail2.sysu.edu.cn

**Keywords:** brain tumors, differentiation therapy, multi-targets therapy, neural regeneration, glioma

## Abstract

Brain tumors have been proved challenging to treat. Here we established a Multi-Target Neural Differentiation (MTND) therapeutic cocktail to achieve effective and safe treatment of brain malignancies by targeting the important hallmarks in brain cancers: poor cell differentiation and compromised cell cycle. In-vitro and in-vivo experiments confirmed the significant therapeutic effect of our MTND therapy. Significantly improved therapeutic effects over current first-line chemo-drugs have been identified in clinical cells, with great inhibition of the growth and migration of tumor cells. Further in-vivo experiments confirmed that sustained MTND treatment showed a 73% reduction of the tumor area. MTND also induced strong expression of phenotypes associated with cell cycle exit/arrest and rapid neural reprograming from clinical glioma cells to glutamatergic and GABAergic expressing cells, which are two key neuronal types involved in many human brain functions, including learning and memory. Collectively, MTND induced multi-targeted genotypic expression changes to achieve direct neural conversion of glioma cells and controlled the cell cycle/tumorigenesis development, helping control tumor cells’ malignant proliferation and making it possible to treat brain malignant tumors effectively and safely. These encouraging results open avenues to developing new therapies for brain malignancies beyond cytotoxic agents, providing more effective medication recommendations with reduced toxicity.

## 1. Introduction

Cytotoxicity-driven tumor cell death is the primary target of most clinical chemotherapeutic strategies; however, it requires sufficient drug concentrations to accumulate in the brain and results in severe systemic side effects with limited efficacy [1,2,3,4,5]. Despite advances [2,6,7] in the development of glioma therapy over the past few decades, the prognosis for brain cancer patients has barely improved. These disappointing results can be partly explained by tumor-developed drug resistance [8,9,10], which is also the reason why those traditional cytotoxic strategies do not achieve and maintain an effective therapeutic state within malignant glioma cells [11,12,13]. As the most widely used oral alkylating agent to treat glioblastoma multiforme (GBM), temozolomide (TMZ) has become a cornerstone of GBM therapy [14,15], but unfortunately it also has serious problems of tumor resistance and poor efficiency [16,17,18,19,20]. It is reported that over 50% of GBM patients treated with TMZ do not respond to the therapy [21,22,23]. Thus, brain tumor treatment remains one of the greatest challenges in oncology. Novel and effective therapeutic strategies are urgently needed.

Poor differentiation is an important hallmark of cancer cells [24,25,26], and reactivating the endogenous differentiation program to restore the maturation process can eliminate the tumor phenotype and holds great promise for cancer treatment. A successful case of differentiation therapy is all-trans-retinoic acid (ATRA) [27,28,29,30], which induces complete remission in patients with acute promyelocytic leukemia (APL). ATRA stimulates the differentiation of immature cancer cells into more mature forms, not only revolutionizing the clinical therapy of APL, but also confirming that inducing terminal differentiation is an effective approach to treat tumors [31,32,33]. Subsequently, emerging studies have focused on exploring the application of differentiation therapy in other types of cancers, especially in solid tumors [34,35,36,37]. Studies have found that targeting RSPO3 in PTPRK-RSPO3-fusion-positive human tumor xenografts promotes cell differentiation and inhibits tumor growth, in part suggesting differentiation therapy as a new clinical approach for the treatment of colorectal tumors [36]. However, the genetic basis of most solid tumors involves the synergistic response to multiple oncogenic mutations and it is therefore much more complex and difficult to achieve effective treatment results from differentiation therapy. In brain cancers, differentiation therapy has also shown a good potential for tumor treatment [38,39]. Researchers found that retinoic acid (RA) can trigger morphological and functional changes in either cell lines or primary patient cells [37,40], which leads to growth arrest, dendrite growth, and the formation of vesicles [37]. In addition, cis-retinoic acid, another isoform of retinoic acid, can improve survival in high-risk neuroblastoma patients by inducing the differentiation of undifferentiated neuroblastoma cells [38,41]. These exciting data raise several interesting possibilities to treat brain tumors and confirm that differentiation therapy holds great promise for the treatment of brain cancers. The differentiation therapy can force self-renewing tumor cells into mature postmitotic cells. Here, we established a promising Multi-Target Neural Differentiation (MTND) therapeutic cocktail to achieve the effective and safe treatment of brain malignancies by targeting the important hallmarks in brain tumors: poor cell differentiation and compromised cell cycle.

## 2. Results and Discussion

### 2.1. Design of MTND

The design principle of the MTND is shown in Figure 1, which targets the important hallmarks in brain cancers—poor cell differentiation and compromised cell cycle—enabling an effective and safe treatment for brain malignances. Cell differentiation or acquisition of specialized cellular functions help drive a malignant tumor to a less invasive state (such as post-mitotic state), and is therefore an attractive chemotherapeutic alternative to suppress tumor growth without the pervasive damage to healthy tissue. The compromised cell cycle found in most cancer cells is the result of mutations that allow cell cycle progression and prevent exit, which is another target of MTND therapy to improve outcomes for brain tumor patients (Figure 1). To suppress tumor persistence and overgrowth, the cell division cycle is directly regulated by MTND-induced multiple cycle control mechanisms, including checkpoint surveillance and proteins involved in delaying or arresting cell cycle progression.

### 2.2. Selection and Mechanism of MTND

Based on two important hallmarks of brain cancers, poor cell differentiation and compromised cell cycle, we proposed a multi-target neural differentiation therapeutic cocktail as a treatment for brain tumors. Through researching the literature, we selected the following eleven small molecules as alternatives, which include Retinoic acid, Forskolin, CHIR99021, Y27632, ISX9, DAPT, PD0325901, Dorsomorphin, LDN193189, Purmorphamine, P7C3-A20. These molecules are mainly involved in cell proliferation, cell migration, neural differentiation, and neuroprotection. We classified all small molecules into the four categories described above. Then one or two small molecules from each of the four categories were selected to form different drug combinations, and the effects of each drug combination were verified in-vitro cell experiments. We selected the final drug combination according to the neural induction efficiency of the drugs. As shown in Figure 2a–c, the induction effect of MTND (10 μM Forskolin, 1 μM Dorsomorphin, 1 μM Purmorphamine, 3 μM CHIR99021 and 3 μM P7C3-A20) therapy was significantly superior to other combinations, so we chose MTND as our final strategy.

The mechanism of the MTND is shown in Figure 2d. The core function of MTND was designed to unblock cellular differentiation in glioma cells through a minimalist three-factor neuronal reprogramming approach, which mainly employed a combined regulation of glycogen synthase kinase-3β (GSK-3β), cyclic adenosine monophosphate (cAMP) and Smad pathways (CHIR99021, Forskolin, Purmorphamine and Dorsomorphin, Figure 2d) to initiate endogenous differentiation of glioma to neurons. A combination of multiple signaling pathways is critical for a successful neuronal conversion from gliomas cells. CHIR99021 is an aminopyrimidine derivative that is an extremely potent glycogen synthase kinase (GSK) 3 inhibitor, inhibiting both GSK3β and GSK3α [42]. GSK-3β is a key molecule in several signaling pathways, including the Wnt/β-catenin signaling pathway, which is involved in the neural differentiation of embryonic, somatic, and neural stem cells [43]. It is reported that inhibition of GSK-3β enhances neural differentiation in unrestricted somatic stem cells [44]. Forskolin is a diterpene adenylate cyclase activator and is commonly used to increase the level of cyclic AMP (cAMP), and has been reported to regulate neuronal specification and promote axonal regeneration [3]. Moreover, as a PKA activator, Forskolin represses cell growth via cell cycle arrest in the G0/G1 phase [45]. The sonic hedgehog (Shh) signaling pathway plays an important role in neurogenesis and neural patterning during development of the CNS, as well as glioblastoma cell proliferation and migration [46,47]. Purmorphamine as a Shh signaling activator can promote neuron differentiation of mesenchymal stem cells [48]. There is a crucial role for SMAD signaling during neural induction, and inhibition of the SMAD pathways by Dorsomorphin drives the differentiation of hESCs to neural lineage [49]. Additionally, AMP-activated protein kinase (AMPK) is an evolutionarily conserved energy sensor important for cell growth, proliferation, survival, and metabolic regulation [50]. Studies have shown that the AMPK inhibitor Dorsomorphin could effectively reduce glioma viability in vitro both by inhibiting proliferation and inducing cell death [8]. P7C3-A20 is a derivative of P7C3 and a highly effective neuroprotective compound that promotes neurogenesis and inhibits the cell death of mature neurons [51]. Through the combined action of the above five molecules, we confirmed the application of MTND in the treatment of brain tumors.

### 2.3. MTND Showing Enhanced Therapeutic Efficacy in In Vitro and In Vivo Experiments

The in vitro therapeutic effects of the MTND were explored on clinical glioma cells harvested through surgical section from three tumor patients in the first affiliated hospital of Sun Yat-sen university. These clinical glioma cells (T-36, T-51, T-59) were treated with the MTND and TMZ, and the corresponding growth and migration were evaluated (Figure 3a,b). EdU staining assay was performed to assess the tumor cell proliferation, while the cell migration was observed and evaluated by wound healing assay. For some patients, both chemotherapy groups presented positive therapeutic effects, with significantly inhibited cell growth (e.g., T-36 patient). However, 2 out of 3 patients (T-51, T-59) showed an obvious resistance to TMZ and no significant inhibited cell proliferation was found in these TMZ treated cells (Figure 3c). Notably, MTND maintained a strong inhibitory effect in all patients (Figure 3c), showing enhanced inhibition of cell growth in all groups, with an improvement of ~78% in T-36 group. Meanwhile, we found that MTND showed significant inhibition of tumor cell migration in all groups, while TMZ made no difference to the migration of all clinical tumor cells (Figure 3d,e). We then explored the therapeutic effects of MTND and TMZ in BALB/c nude mice bearing clinical human glioma tumor xenografts of similar size. MTND and TMZ chemo-drugs were applied daily through a brain stereotaxic tube on top of freely moving mice (Figure 3f), and the tumor burden was analyzed by histological studies after 9 days of treatment. As shown in Figure 3g,h, the MTND-treated group developed much smaller tumors in the mouse brains compared to the TMZ and SHAM controls. These in vitro and in vivo findings confirm that MTND therapy has higher therapeutic efficacy than traditional treatment regiments.

### 2.4. MTND Induces Multi-Target Therapeutic Genotype Alterations to Achieve Cell Cycle Exit/Arrest

To verify the effect of MTND on inducing cell cycle arrest, we performed RNA-seq on MTND-treated cells (T-26). We found that MTND induces multiple genotypic alternations in glioma, such as glioma pathogenesis and neural differentiation. As shown in the Venn diagram of differentially expressed genes (Figure 4a), distinct changes in gene expression in response to MTND treatment was observed, and additional variation increased substantially (doubled) with increasing treatment time. We classified the significant regulated biological pathways into four categories as seen in Figure 4b, including glioma pathogenesis, cell growth and death, neural differentiation, and neural maturation. These modulations not only ameliorated glioma pathogenesis, directly inhibited glioma malignant proliferation, and controlled the cell cycle, but also activated neural differentiation at the same time. Enhanced synaptic vesicle cycling and axon guidance/regeneration indicated a neural maturation process during MTND treatment, which was further confirmed by protein staining and cell morphology. Analysis of the first few up/down-regulated genes confirmed the important role of MTND in ameliorating glioma pathogenesis and promoting neural differentiation (Figure 4c). Multiple genes related to glioma tumorigenesis and progression were targeted and down-regulated by the MTND, including SGK3, NLPRP10 and OXTR. Meanwhile, most of the top up-regulated genes were involved in WNT/BMP signaling modulation and induced neural differentiation, including NKD1, APCDD1, LAMP5, and NOTUM.

RNA-based measurements indicated a strongly expressed phenotype of cell cycle exit/arrest during MTND therapy (Figure 4d, green for down-regulation and red for up-regulation), which is one of the major mechanisms for inhibiting tumor progression. Several upstream pathways involved in the regulation of the cell cycle (such as PI3K and MAPK, Figure 4d) were found to be down-regulated after MTND treatment, resulting in hindered commitment to replication initiation and S phase entry. PI3K-Akt signaling pathway is commonly dysregulated in a variety of human tumors. Function gain or loss caused by certain gene mutations in this pathway can lead to cell transformation and the proliferation and survival of tumor cells, and is closely related to tumor invasion and metastasis. Akt mediates cell proliferation and invasion in many tumors. By down-regulating the PI3K-Akt signaling pathway, MTND can arrest tumor cell cycle, thereby inhibiting tumor cells’ proliferation and migration. In addition, permanent activation of Ras proteins by mutations is common in all human tumors, so inhibitors targeting Ras are effective agents for cancer treatment. MTND controls cell cycle checkpoints by down-regulating the Ras/Raf/MEK signaling pathway, thereby achieving cell cycle exit.

As the main driver of cell cycle progression, the Cyclin-CDK complex is altered in nearly 80% of human gliomas and is one of the three most perturbed pathways [52,53]. By regulating Cyclin–CDKs activity (Figure 4e), MTND arrests cell cycle in a prolonged pre-S phase and induces a cell cycle exit known as quiescence. In addition, overexpression CDK7 was found to be corrected (Figure 4e), which reduces CDK7-induced phosphorylation of the main cell cycle-dependent CDKs and could reverse the poor prognosis of GBM [54,55]. We further identified the down-regulation of oncogenes (EGFR and PDGFB) and up-regulation of tumor suppressors (PTEN and GADD45A) after MTND treatment (Figure 4f), which are closely associated with the pathogenesis of glioma proliferation. EGFR/PDGFB are overexpressed in most primary glioblastomas, resulting in hyperactivation of cell proliferation and invasion. Abundant molecular alternations inactivate tumor suppressors, further causing uncontrolled cell proliferation of glioma, which are critical for glioma pathogenesis [56,57,58,59]. It is worth noting that although the negative regulator MDM2 was down-regulated, p53 expression was abnormally decreased after treatment. In addition to p53, several other aberrantly regulatory factors, such as Rb and CDK2/4, did not exhibit the expected responses from MTND treatment, suggesting that uncontrolled cell cycle pathogenic mutations may contribute to tumor therapy resistance. The complexity of glioma pathogenesis is a major reason for the limited efficacy and rapid acquisition of tumor resistance, and our multi-targeted MTND approach successfully suppressed malignant tumor proliferation through alternative targets or by bypassing these uncontrolled abnormal mutations. Collectively, the MTND induces multi-targeted genotypic expression changes to achieve the direct neural conversion of glioma cells and inhibits the cell cycle/tumorigenesis development, helping control tumor cells’ malignant proliferation.

### 2.5. MTND Facilitates Neural Regeneration

Combined action of MTND is sufficient to induce rapid neural reprograming from clinical glioma cells to neural type cells, which was confirmed on clinical glioma cells (T-26) by immunostaining of neural proteins (Figure 5a). Before MTND treatment, neuron-specific class III beta-tubulin (TUJ1) immunoreactivity was barely observed. After two days of MTND treatment, the cultures turned into TUJ1 positive with short unipolar, bipolar, or elongated processes, arborized neurites, and occasional pyramidal morphologies (Figure 5a). When quantified after 5 days’ treatment, these neuron-like phenotypes accounted for ~40% the total cell population and the expression of glioma-associated nestin gradually disappeared (Figure 5b). Nestin is strongly expressed in GBM and in most IDH wild-type gliomas. Studies showed that Nestin expression and intensity were significantly associated with poor survival, and Nestin defects in GBM and U251 cells can inhibit the growth of transplanted tumors. What is more, the disappearance of Nestin, a potential marker of cancer stem cells, indicated that those treated cells did not possess stem cell characteristics which potentially contributed to tumor relapse due to their tumor-initiating capabilities. We then found that in the gene-expression profile of MTND-induced cells, functional categories associated with neurogenesis, synaptic transmission, and axon genesis were upregulated (Figure 5c). Increased expression of vesicular glutamate transporter 1 (VGLUT1) implies the increase of glutamatergic neurons, which would lead to an increase of the most common excitatory neurotransmitters in the central nervous system (CNS). Meanwhile, GABA transporter 1 (GAT1), which is involved in transporting the primary inhibitory neurotransmitter in the cerebral cortex, was also significantly elevated after MTND treatment. These two neuron types are involved in a variety of human brain functions, including learning and memory, suggesting a potentially modulated neural function from MTND treatment in the damaged neural networks.

### 2.6. MTND Induces Low Resistance Development and Lower Toxicity Stress

Consistent exposure to chemotherapeutic agents promotes drug resistance in cancer cells, which is the major obstacle to successful cancer suppression. To evaluate the drug resistance induced by TMZ and MTND treatment, we preformed long-term recordings of inhibition of cell proliferation in cells T-26. As shown in Figure 6a, by comparing the two reagents, we can see that MTND still has an inhibitory effect on cell proliferation after 8 consecutive days of treatment, while TMZ has initially shown an upward trend in cell proliferation. Therefore, we deduced that MTND could maintain a higher therapeutic effect than TMZ after continuous treatment without inducing significant cell resistance. That is to say, MTND induced low drug resistance and had a lasting therapeutic effect.

One of the reasons for the lower drug resistance to MTND therapy is due to its lower cytotoxicity, which reduces the rapid mutation of tumor cells from the toxic stress. We performed a cell apoptosis/necrosis analysis to verify the low cytotoxic pressure from MTND by exploring phosphatidylserine (PS) residues (normally hidden within the plasma membrane) on the cell surface and characterizing the loss of the integrity of the plasma by propidium iodide (PI) (Figure 6b,c). Compared to the control group, a significantly increased externalized PS and ruptured membrane were identified in TMZ-treated cells, whereas MTND-treated cells exhibited a similar apoptosis/necrosis level as the control group, indicating a negligible extra toxic stress from MTND groups.

## 3. Materials and Methods

### 3.1. Cell Culture

All the research procedures were approved by the ethics committee of Sun Yat-sen University (SYSU-IACUC-2021-B0989), and informed consent was obtained from all patients. Human clinical glioma tumors were resected through surgery and collected for use. Tumor and surrounding cells were processed and separated by mechanical trituration, trypsinization, filtration, and centrifugation. Cells were cultured in Dulbecco’s Modified Eagle Medium/Nutrient Mixture F-12 (Hyclone, Logan, UT, USA) containing 10% fetal bovine serum (Hyclone, Logan, UT, USA), 2.5 mmol/L L-glutamine, and 100 U/mL penicillin–streptomycin solutions (Gibco, Grand Island, NY, USA). Cells were incubated at 37 °C in a humidified atmosphere with 5% CO_2_ and medium change every other day. The number of cell passages used in both in vitro and in vivo experiments were passage 2–5.

### 3.2. MTND Treatment

The tumor cells were cultured on poly-D-lysine-(Sigma, St. Louis, MO, USA) and laminin-(Sigma, St. Louis, MO, USA) coated coverslips (9 mm) at a density of 5000 cells per coverslip. The cells were cultured in DMEM/F12 containing MTND (10 μM Forskolin, 1 μM Dorsomorphin, 1 μM Purmorphamine, 3 μM CHIR99021 and 3 μM P7C3-A20) or 0.1% DMSO in control groups. Half the medium was refreshed every day and MTND was given once daily. All small molecules were purchased form Glpbio (Montclair, CA, USA).

### 3.3. Immunofluorescence Staining

Samples (T-26) were fixed for 30 min in 4% paraformaldehyde (Biosharp, Hefei, China), permeabilized in 0.25% Triton X-100 for 15 min, and then blocked with 4% BSA for 2 h at room temperature or overnight at 4 °C. Then, samples were incubated with primary antibodies for 4 h at room temperature followed with secondary antibodies for 2 h. Images were captured with a fluorescence microscope (Cnoptec, Chongqing, China). The primary antibodies used were: Tuj1 (4466S, Cell Signaling Technology, Boston, MA, USA), Nestin (ab105389, Abcam, Cambridge, UK). The secondary antibodies used were: Anti-rabbit IgG (H + L) (Alexa Fluor^®^ 594 Conjugate, 8889, Cell Signaling Technology, Boston, MA, USA), anti-mouse IgG (H + L) (Alexa Fluor^®^ 488 Conjugate, 4408, Cell Signaling Technology, Boston, MA, USA).

### 3.4. EdU Cell Proliferation Assays

Cell proliferation assay was performed using an EdU kit (BeyoClick™ EdU Cell Proliferation Kit with Alexa Fluor 488, Beyotime, Shanghai, China). Briefly, Cells (T-36, T-51, T-59) seeded on coverslips were cultured in DMEM/F12 medium with treatment (0.1% DMSO, 200 μm TMZ or MTND). A total of 48 h later, cells were incubated with 10 μM EdU for 24 h, followed by fixing with 4% paraformaldehyde (PFA) for 15 min and permeating with 0.3% Triton X-100 for another 15 min at room temperature (RT). Subsequently, the samples were incubated with the Click Reaction Mixture at RT for 30 min in the dark and then incubated with Hoechst 33,342 for 10 min. Images were captured with a fluorescence microscope (Cnoptec, Chongqing, China). The improvement in the cell proliferation inhibition rate of MTND compared with TMZ group was calculated according the formula: (Proliferation rate_(TMZ)_-Proliferation rate_(MTND)_)/(Proliferation rate_(Ctrl)_-Proliferation rate_(TMZ)_).

### 3.5. Cell Migration Assays

Wound-healing assay was applied to assess cell migration. An appropriate number of cells (T-36, T-51, T-59) were plated in a 6-well plate for 100% confluence in 24 h. Clear lines were created with a sterile 200μL pipette tip. After washing with medium to remove cell debris, the cells were continuously cultured in the serum-free medium to minimize cell proliferation and treated with 0.1% DMSO, 200 μm TMZ or MTND. The wound areas were recorded at 0 h, 24 h, and 48 h using a microscope (Cnoptec, Chongqing, China), and the migration distance was measured by the ImageJ software (1.53q).

### 3.6. In-Vivo Experiments

All procedures involving animals were approved by the Animal Ethical Committee of Sun Yat-sen University (SYSU-IACUC-2021-B0989). The immunodeficient BALB/c nude mice were purchased and housed in Sun Yat-sen University Experiment Animal Center under a 12 h light/dark cycle and had ad libitum access to food and water. Nine BALB/c nude mice were divided into three groups with three mice in each group. All of them were injected with clinical tumor cells (T-36) to construct an orthotopic transplantation tumor model. The experimental group were injected with TMZ and MTND, respectively. And the control group was injected with normal saline. Glioma orthotopic transplantation tumor model was created by injecting clinical cells in lateral ventricles and similar deficient ones were selected for following experiments. Mice were anesthetized with 5% chloral hydrate (intraperitoneal, 0.08 mL/10 g) to relieve pain during the surgical procedure. First, the surgical area of the mouse head was sterilized with a cotton ball containing 75% medical alcohol, and then we cut the scalp along the midline of the mouse’s scalp. A 0.6 mm diameter hole in the skull was drilled. Clinical gliomas cells (5 × 10^5^ cells in 5 μL) were implanted into the right hemisphere perpendicularly at a location 1.5 mm posterior to the bregma, 1.5 mm lateral to the sagittal suture, and at a 3.0 mm depth from the skull surface. After the needle was removed, the brain stereotactic tube was fixed to the mouse skull with dental cement under the same stereotactic coordinates along the needle track for subsequent medicinal use. The incision was then sutured. Subsequently, drugs were injected in the freely moving mice’s brains with a brain stereotaxic tube under the same stereotactic coordinates along the needle track. After 10 days of treatments, mice were anesthetized with chloral hydrate and perfused with normal saline and following 4% PFA. At the end of perfusion, brain tissues were removed by craniotomy and fixed in 4% PFA for histological sections. Before HE staining, the entire brains of the mice were sliced along the coronal plane at regular intervals (40 μm) to ensure that the tumor in the brain was completely sectioned and made into brain slices. Then, the brain slice with the largest tumor area was selected as the tumor area size of each mouse to ensure the scientific validity of our results.

### 3.7. RNA Sequencing and Data Analysis

Total RNA samples were extracted from clinical glioma cells (T-26) using Trizol Reagent (Thermo Fisher Scientific) according to the manufacturer’s instructions. Total RNA concentration, RIN value, 28S/18S and fragment size were determined on the Agilent Technologies 2100 bioanalyzer. RNA sequencing is performed by BGISEQ platform (BGI Genomics Co., Ltd., Shenzhen, China). For insight into the change of phenotype, GO- (http://www.geneontology.org/ (accessed on 22 February 2022)) and KEGG- (https://www.kegg.jp/ (accessed on 22 February 2022)) enrichment analysis of annotated different expression gene was explored. Pathway analysis: To identify biological pathways that are differentially regulated after Day2/5 MTND treatment, we analyzed the change of related functional pathways based on KEGG (Kyoto Encyclopedia of Genes and Genomes) database. We analyzed the change of gene expression relating to MTND treatment. The signaling pathways that are activated in cells after exposure of MTND-conditioned medium were evaluated by sequencing RNA. The significant levels of terms and pathways were corrected by *Q* value with a rigorous threshold (*Q* value ≤ 0.05) by Bonferroni.

### 3.8. Cell Apoptosis Assays

Cell apoptosis assay was performed using Annexin V-FITC/Propidium Iodide (PI) Apoptosis Detection Kit (Keygen Biotech, Nanjing, China). After treatments (0.1% DMSO, 200 μm TMZ or MTND) for 48 h, T-26 cells seeded on coverslips were washed twice with 0.01 M PBS and incubated with Hoechst 33,342, Annexin V-FITC and PI at RT for 10 min. Fluorescence images were immediately observed under fluorescence microscope (Cnoptec, Chongqing, China).

### 3.9. Information of Clinical Cell Lines

Cells for all the tests were from clinical glioma patients with detailed information as follows:T-26;WHO grade: III;Histology: Anaplastic astrocytoma;Molecular background: IDH1(-); MGMT gene promoter methylation(+); PTEN, TSC2 deletion; CDK4, CDK6, EGFR, MET, SMO, MDM4, TERT amplification; FGFR3–TACC3, PIK3CAp.K111E, PTENp.I101T, BCORp.A606fs cancer driver genes mutation;Prior treatment with radiation: There was no radiotherapy before the tumor was surgically removed.T-36;WHO grade: IV;Histology: Glioblastoma;Molecular background: IDH1(-); TERT promoter region mutation; MGMT gene promoter methylation(+); BRCA2, CDKN2A, CDKN2B, PTEN deletion; CDK6, SMO amplification, PIK3CAp.H1047Y, STAG2p.E470 cancer driver gene mutation;Prior treatment with radiation: There was no radiotherapy before the tumor was surgically removed.T-51;WHO grade: I;Histology: Meningiomas;Molecular background: NF2p.A451fs cancer driver gene mutation;Prior treatment with radiation: There was no radiotherapy before the tumor was surgically removed.T-59;WHO grade: IV;Histology: Glioblastoma;Molecular background: IDH1(-); TERT promoter region mutation; MGMT gene promoter methylation(+); SMO amplification; TP53p.R280K, BRCA2p.N289H, CARD11p.K83M, NOTCH2p.H107P cancer driver genes mutation;Prior treatment with radiation: There was no radiotherapy before the tumor was surgically removed.

## 4. Conclusions

The presented work demonstrates a novel approach, MTND therapy, to achieve the effective and safe treatment of brain malignances through unblocking their endogenous differentiation. In vitro and in vivo experiments confirmed the significantly improved therapeutic effect of MTND compared to temozolomide, significantly inhibiting the proliferation and migration of brain tumors. Our therapy can achieve the effective and safe treatment of brain tumors through unblocking their endogenous differentiation. We found that MTND can induce the rapid and efficient transformation of clinical glioma cells into neurotype cells, including glutamatergic- and GABAergic-expressing cells, which are two key neuronal types involved in many human brain functions, including learning and memory. These results support the possibility of using MTND to develop therapeutics for neurological diseases that affect learning and memory. In addition, the RNA-seq results indicated that MTND induced strong expression of phenotypes associated with cell cycle exit/arrest and multi-targeted genotypic expression changes. Several upstream pathways involved in the regulation of the cell cycle (such as PI3K and MAPK) were found to be down-regulated after MTND treatment, resulting in hindered commitment to replication initiation and S phase entry. These regulations can not only improve the pathogenesis of glioma, control the cell cycle, and directly inhibit the malignant proliferation of glioma, but also promote neural differentiation, further validating the effectiveness of our Multi-Target Neural Differentiation therapy. More importantly, MTND therapy showed little resistance due to low toxic stress and maintained a more effective and long-lasting therapeutic effect than TMZ. Collectively, MTND therapy induced multiple-target genotype expression changes to achieve the direct neural transformation of glioma cells and inhibit cell cycle/tumorigenesis progression, which contributed to suppress the malignant proliferation of tumor cells and had the potential to modulate impaired neurological function. These encouraging results open avenues to developing new therapies for brain malignancies beyond cytotoxic agents, and may change the current landscape of tumor treatment.

## Figures and Tables

**Figure 1 ijms-24-12329-f001:**
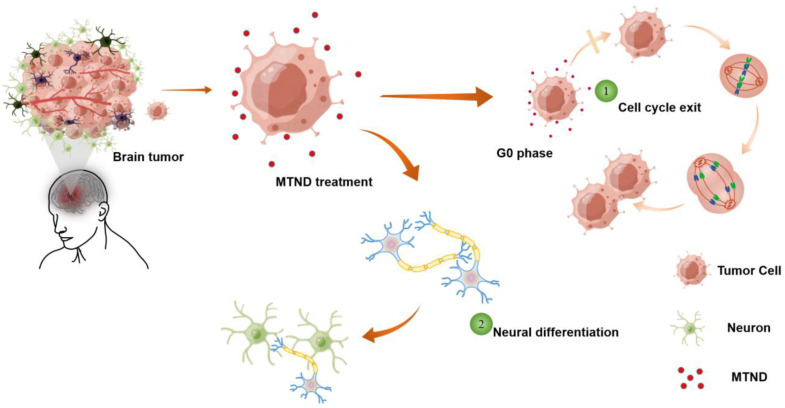
Schematic of the MTND therapy.

**Figure 2 ijms-24-12329-f002:**
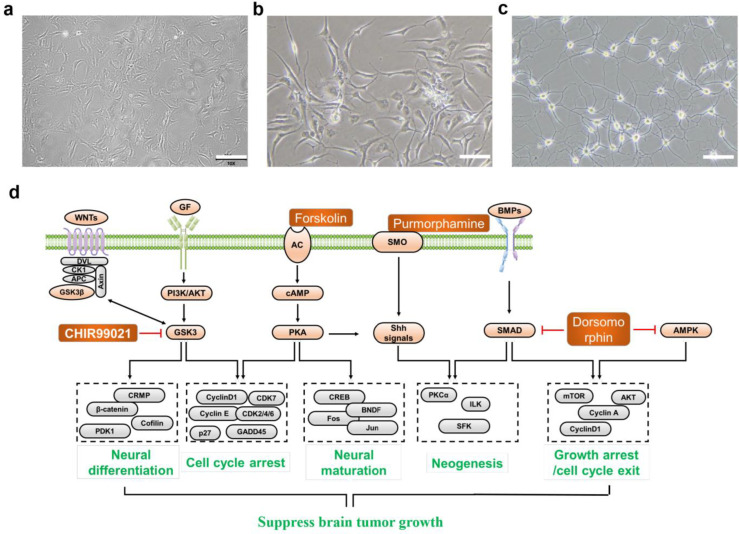
Selection and mechanism of the MTND therapy. (**a**) The cell morphology (bright field image) of tumor cells in its normal state (without any drug treatment); (**b**) tumor cell morphology (bright field image) under other drug combinations (Retinoic acid, Dorsomorphin, Purmorphamine, P7C3-A20); (**c**) tumor cell morphology (bright field image) under MTND treatment. Scale bar: 200 μm. (**d**) Mechanism of the functions of MTND for brain tumor treatment.

**Figure 3 ijms-24-12329-f003:**
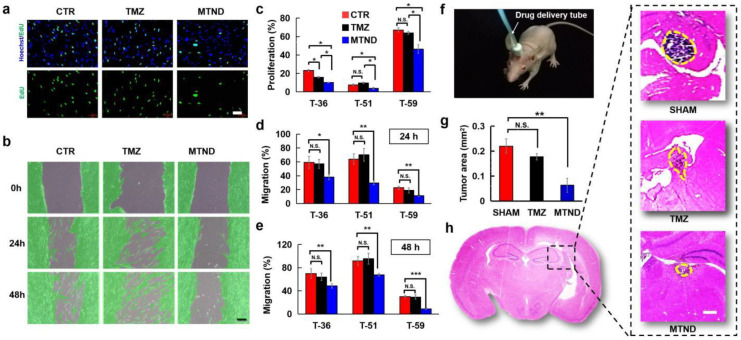
Enhanced therapeutic efficacy was found after MTND treatment. (**a**) EdU cell proliferation assay of clinical brain tumor cells (T-36). Green: EdU; blue: Hoechst. Scale bar: 100 μm. (**b**) Cell migration (T-36) detected by the wound-healing assay with cells shadowed in green. Scale bar: 100 μm. (**c**) Statistical results of cell proliferation at 72 h. The percentage represents the proportion of newly divided cells in the total cell population, and 100% indicates that all the cells on the coverslips were newly proliferated during the test. (**d**,**e**) Statistical results of cell migration at 24 h and 48 h. The percentages represent the ratio of the area of cell migration to the area of the initial scratch area, and 100% indicates that the cells have completely covered the scratched area. (**f**) A photo of a brain stereotaxic tube on top of freely moving mice for drug delivery. (**g**,**h**) Histological analysis of brain tumor (T-36) shows that the MTND-treated group developed much smaller tumors in the mice brains compared to the TMZ and SHAM controls. Scale bar: 100 μm; N.S.: not significant; *: *p*-value < 0.05; **: *p*-value < 0.01; ***: *p*-value < 0.001.

**Figure 4 ijms-24-12329-f004:**
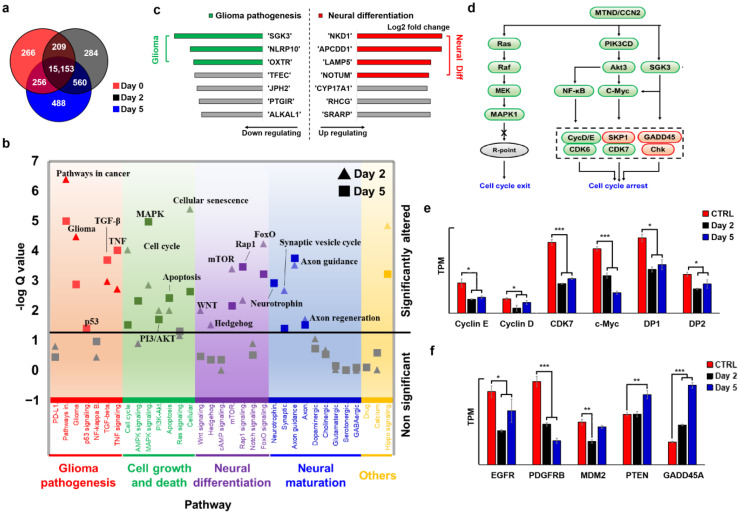
MTND induces and multi-targets therapeutic genotype alterations to achieve cell cycle exit/arrest. (**a**) Venn diagram of differentially expressed genes after MTND treatment. (**b**) The significant MTND-regulated biological pathways were classified into 4 categories: glioma pathogenesis, cell growth and death, neural differentiation, and neural maturation. (**c**) Top 7 up/down-regulated genes after MTND treatment. (**d**) A strongly expressed phenotype of cell cycle exit/arrest was identified after MTND treatment. Green for down-regulation and red for up-regulation. (**e**) RNA expression of genes related cell cycle. (**f**) Down-regulation of oncogenes (EGFR and PDGFB) and up-regulation of tumor suppressors (PTEN and GADD45A) were identified after MTND treatment. *: *p*-value < 0.05; **: *p*-value < 0.01; ***: *p*-value < 0.001.

**Figure 5 ijms-24-12329-f005:**
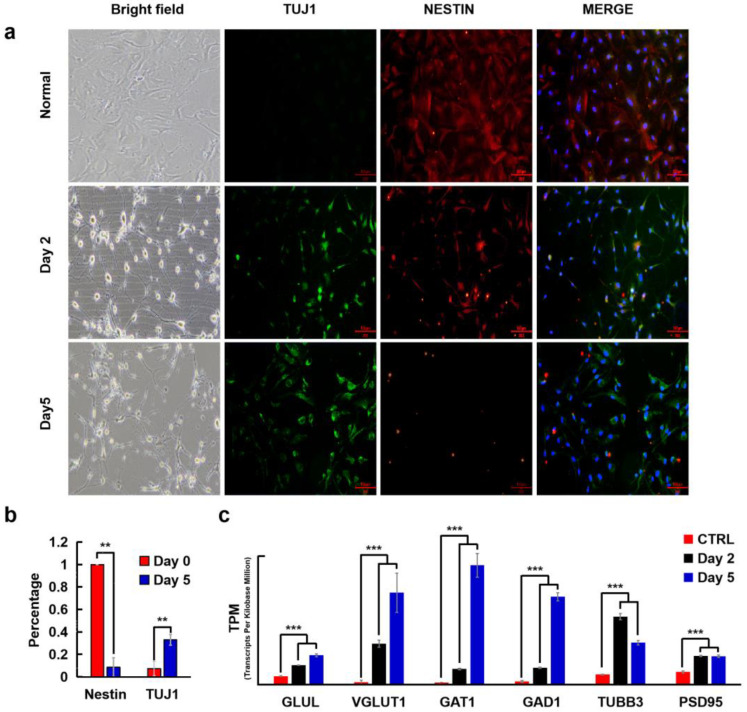
MTND facilitates neural regeneration. (**a**) Bright field images and immunostaining images of clinical tumor cells (T-36) after MTND treatment. Scale bar: 100 μm. (**b**) Statistical analysis of TUJ1 and Nestin percentages after 5-day MTND treatment. (**c**) Functional categories associated with neurogenesis, synaptic transmission and axon genesis were upregulated after MTND treatment. **: *p*-value < 0.01; ***: *p*-value < 0.001.

**Figure 6 ijms-24-12329-f006:**
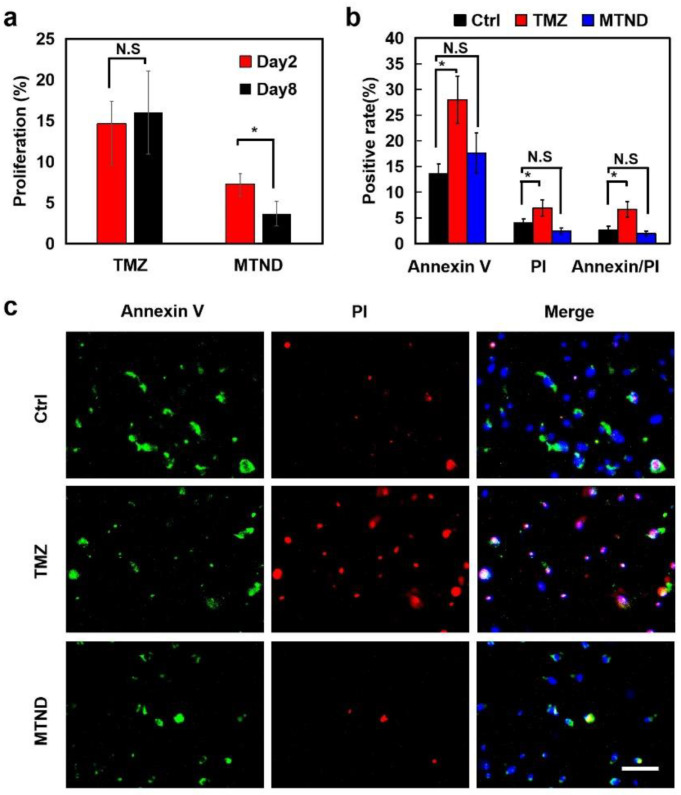
MTND induces low resistance development and lower toxicity stress. (**a**) Increased cell proliferation was found in TMZ-treated glioma cells after 8 days of consistent treatment, while no similar drug resistance was observed in the MTND group. (**b**,**c**) Compared to control group, a significantly increased externalized PS and ruptured membrane were identified in TMZ-treated cells, whereas MTND-treated cells exhibited a similar apoptosis/necrosis level as control group. Scale bar: 100 μm. N.S.: not significant; *: *p*-value < 0.05.

## Data Availability

The data that support the findings of this study are available upon reasonable request.

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
