# Peer review of "Multi-Target Neural Differentiation (MTND) Therapeutic Cocktail to Suppress Brain Tumor"

_ijms, 2023, doi:10.3390/ijms241512329_

Round 1

Reviewer 1 Report (Previous Reviewer 4)

Currently, the critical point of target therapy is represented by the pleiotropism of tumor cells, which can simultaneously express different downstream or upstream markers of a specific molecular pathway. For this reason, especially for brain tumors such as glioblastoma (GBM), where inter- and intra-tumor heterogeneity prevails, the present study may assume considerable importance. The authors of this work tested a multi-target neuronal differentiation (MTND) therapy with the main aim of addressing GBM cell anaplasia by promoting neuronal differentiation. In the first part the authors performed an evaluation of the MTND treatment on cell viability and migration, compared to the standard treatment with temozolomide, both in vitro and in vivo. Then, the authors performed an analysis of RNA sequencing after treatment with MTND, reporting the activation of pathways correlated with neuronal differentiation and with the inhibition of tumor growth. In conclusion, the authors demonstrate that MTND is associated with improvements in learning and memory functions attributable to reprogramming of neuronal differentiation. These results may therefore be promising also for a therapy that avoids side effects.

 Broad comments 

The study addresses a problem of great interest to the scientific community for the production of drugs with a broad spectrum in the field of molecularly targeted drugs. The authors explain how molecules and factors can act on the neuronal cell, correctly reprogramming the cell cycle and directing them to cell differentiation. The study shows that MTND therapy can open up interesting perspectives in solving these problems.

 Specific comments

The authors resubmitted the new version of the paper which I had previously judged as accept after minor revision. In this new version, the authors have solved the critical issues I raised.

The Language of this manuscript has been improved with short and simple and sentences.

Author Response

We thank the reviewers for the positive comments.

Reviewer 2 Report (New Reviewer)

1. The discussion of the effect of MTND on animal behavior detracts from the presentation of the effect of MTND on the tumors.  Please remove it from this manuscript and use the data for another manuscript.

2. The paper is well-written overall but there are many grammatical errors.  The following changes are suggested:

line 17 "greatly" should be "great"

line 21 "shown" should be "showed"

line 27-28 "providing a more efficacy" should be "providing more effective..."

line 34 should be " ; however" the punctuation should be a semi-colon not a comma

line 45 should be "oncology.  Novel..."

line 54 "has" should be "have"

line 58 "clinically" should be "clinical"

line 60 should be "it is"

line 61 should be "treatment results"

line 70 Remove "to treat tumor"

line 71 Remove "Here"; duplication

line 103 This is not a sentence.  Do you mean to say "All are classified as small molecules"?

line 121 should say "which is involved"

line 197 should say "related to"

line 218 should say "arrests"

line 220 "reducing" should be "reduces"

line 221 "could be relieve" should be "could reverse the poor prognosis of GBM"

line 225 change to " resulting in hyperactivation of..."

line 226 should be " invasion.  Abundant..."

line 230 "regulation" should be "regulatory factors"; remove "which"

line 235 should be "or by bypassing"

line 261  should be "and in most IDH"

line 263 check meaning: Do you mean "can inhibit"?

line 264 change to "disappearance of Nestin as a potential..."

line 265 change "otentially" to "potentially"

line 325 remove "developed"

line 327 change "chem-induced" to "induced" or another word of your choosing

line 527 in the conclusion: change "are expected ..." to "may" to avoid overreaching in the conclusion.

There are many grammatical errors in the text; however, overall it is well-written.

Author Response

We appreciate the time and effort that you and the reviewers dedicated to providing feedback on our manuscript and are grateful for the insightful comments on and valuable improvements to our paper. Our response is given in normal font and changes/additions to the manuscript are given in the blue text.

Reviewer2

1.The discussion of the effect of MTND on animal behavior detracts from the presentation of the effect of MTND on the tumors.  Please remove it from this manuscript and use the data for another manuscript.

The author’s answer: Thanks for your suggestion. We have removed this part according to the Reviewer’s suggestion.

  1. The paper is well-written overall but there are many grammatical errors.  The following changes are suggested:

line 17 "greatly" should be "great"

line 21 "shown" should be "showed"

line 27-28 "providing a more efficacy" should be "providing more effective..."

line 34 should be " ; however" the punctuation should be a semi-colon not a comma

line 45 should be "oncology.  Novel..."

line 54 "has" should be "have"

line 58 "clinically" should be "clinical"

line 60 should be "it is"

line 61 should be "treatment results"

line 70 Remove "to treat tumor"

line 71 Remove "Here"; duplication

line 103 This is not a sentence.  Do you mean to say "All are classified as small molecules"?

line 121 should say "which is involved"

line 197 should say "related to"

line 218 should say "arrests"

line 220 "reducing" should be "reduces"

line 221 "could be relieve" should be "could reverse the poor prognosis of GBM"

line 225 change to " resulting in hyperactivation of..."

line 226 should be " invasion.  Abundant..."

line 230 "regulation" should be "regulatory factors"; remove "which"

line 235 should be "or by bypassing"

line 261  should be "and in most IDH"

line 263 check meaning: Do you mean "can inhibit"?

line 264 change to "disappearance of Nestin as a potential..."

line 265 change "otentially" to "potentially"

line 325 remove "developed"

line 327 change "chem-induced" to "induced" or another word of your choosing

line 527 in the conclusion: change "are expected ..." to "may" to avoid overreaching in the conclusion.

The author’s answer: Thank you for pointing these out and we have made modifications to the manuscript according to the Reviewer’s suggestion. Changes/additions to the manuscript are given in the blue text.

Round 2

Reviewer 2 Report (New Reviewer)

The authors have made the changes requested.

On the line 262: Should be "contributed"

This manuscript is a resubmission of an earlier submission. The following is a list of the peer review reports and author responses from that submission.

Round 1

Reviewer 1 Report

In this study authors described results from a new therapeutic approach aiming at blocking brain tumor proliferation with a multi-target neuronal differentiation cocktail. Results contain data from in vitro and in vivo experiments, and show the effects of the therapy on glioma phenotype and on mice brain functions.

The results are interesting and contribute to promote a promising field of research. However, conclusions are not completely supported by data.

Major points

-further details are necessary about cell culture. Authors should confirm that genotype of the in vitro culture matches the molecular background of the original patient.  The number of culture passages should be always indicated in the details of the experiments. In some figures it is not clear if results are referred to one or more cell line. T26 are cited in M&M but no results from these cells are visible in figures.

-in Fig 2(b, d, e) data are expressed as percentage but it is not clear which experimental point represents the 100%. Authors should explain in the legend.

-The histology analysis in figure 2H could be affected by bias. Authors should explain in M&M how the maximum diameter of tumor mass was selected.

Author Response

We sincerely thank the editor and all reviewers for their valuable feedback that we have used to improve the quality of our manuscript. The reviewer comments are laid out below in italicized font and specific concerns have been numbered. Our response is given in normal font and changes/additions to the manuscript are given in the blue text.

Reviewer1

Comments and Suggestions for Authors

In this study authors described results from a new therapeutic approach aiming at blocking brain tumor proliferation with a multi-target neuronal differentiation cocktail. Results contain data from in vitro and in vivo experiments, and show the effects of the therapy on glioma phenotype and on mice brain functions. The results are interesting and contribute to promote a promising field of research. However, conclusions are not completely supported by data.

Major points

  1. -further details are necessary about cell culture. Authors should confirm that genotype of the in vitro culture matches the molecular background of the original patient. The number of culture passages should be always indicated in the details of the experiments. In some figures it is not clear if results are referred to one or more cell line. T26 are cited in M&M but no results from these cells are visible in figures.

The author’s answer: Further details about cell culture have been added to M&M. We sequenced and compared about 27,000 coding genes in the whole exome of tumor tissues and normal tissues of patients to obtain the specific variation information in tumors. We performed RNA-seq on patient-derived cultured cells in vitro to analyze the difference in gene expression before and after treatment, so we did not perform RNA-seq on cells from normal tissues. However, we try our best to conduct scientific culture and experiments with appropriate culture passage number to ensure the stability of genotype of the in vitro culture. The number of cell passages in this manuscript were passage 2-5. In this manuscript, T-26 were used for RNA-seq, immunofluorescence staining experiment, drug resistance experiment, and cell apoptosis assays, while T-36 were used for in vivo experiments. Specific details above have been added to M&M.

  1. -in Fig 2(b, d, e) data are expressed as percentage but it is not clear which experimental point represents the 100%. Authors should explain in the legend.

The author’s answer: The percentage in Fig 2c represents the proportion of newly divided cells in the total cell population during EdU test, and 100% indicates that all the cells on the coverslips were newly proliferated during the test. The percentages in Fig 2(d, e) represent the ratio of the area of cell migration to the area of the initial scratch area during cell migration assays, and 100% indicates that the cells have completely covered the scratched area. We have made the above additions. Please see page 4, lines 126–130 of the revised manuscript.

  1. -The histology analysis in figure 2H could be affected by bias. Authors should explain in M&M how the maximum diameter of tumor mass was selected.

The author’s answer: The entire brain of the mouse was sliced along the coronal plane at regular intervals (40μm) to ensure that the tumor in the brain was completely sectioned and made into brain slices. Then, the brain slice with the largest tumor area was taken from all the brain slices as the tumor area size of mouse to ensure the scientific of our results. The above expression have been added to M&M. Please see page 11, lines 368–372 of the revised manuscript.

Reviewer 2 Report

Overall comments

It goes without saying that glioma is one of the diseases that urgently need better treatment. The authors of this paper describe a drug combination they used to induce differentiation in glioma and compare it to commonly used drug – Temozolomide. Unfortunately, the paper is very poorly organized, it’s very hard to understand what was done, why and on what samples. First of all, we don’t know how did the authors select drugs for this combinatorial treatment and how were the drug concentrations established. The first figure shows some in vitro and some in vivo results, both are insufficient to draw any conclusions. Then the authors jump to RNA sequencing results, however they don’t say what cells were analysed there (from patients or cultured or from mice?). Later on it is still very confusing, unfortunately I was too exhausted to try making sense out of it.

Abstract

The abstract is not following the standard plan: introduction, methods, results, conclusions. It’s hard to say what are the results of this study and what was published before. Authors start writing about the results in the 3rd sentence. Please, re-write it in a clearer way. Also, be more concise as at the moment the abstract is over 250 words, which is too long.

Introduction

43-46 – please re-write this sentence to make it grammatically and logically correct

The authors forgot to mention that another isoform of retinoic acid, 13-cis RA, is used to treat neuroblastoma (which is a pediatric solid tumor) in patients with minimal residual disease.

Results and discussion

Section 2.1 - design and mechanism of MTND should be moved to Introduction, it is not a result. Also, please explain whether this MTND is something the authors established or is it widely used. How were the drugs selected and the concentrations optimized? The mechanism is only described in Figure 1b, but should be more detailed also in the text of the manuscript as this is the reason for all experiments.

Figure 1 – a- this figure suggests cells are extracted from patient’s tumor, treated ex vivo and then transferred back to the patient. Based on the text I don’t think that’s the case, so please change it to be less confusing. Also, from this figure it looks like MTND is blocking cell proliferation which leads to cell differentiation. But lack of proliferation does not equal differentiation induction, so please be more specific.

Section 2.2 – it’s crucial to explain what drug concentrations were used and why (how were they established, especially for drug combinations). Were these primary cell lines, or were they established earlier? You cannot say “more than half of the patients” if you only tested 3 patients (better say 2 out of 3). Where did this number 350% come from? I don’t see it anywhere on the graphs and authors are referring to Figure 2c, which is a would healing assay that is not including any numbers. Both text and figure is very confusing, the authors should read it carefully and correct it. What is a SHAM control?

Line 110 – what do the authors mean by morphology studies? Were the mice euthanized and then tumor burden was analyzed on histological sections?

Figure 2 – a-which cell line is it? The labels in the text do not match those in the figure (b mixed with c, etc). g – how many mice are represented by each bar?

To me, it’s not understandable why the authors jumped from 2 in vitro experiments to an animal study. There is definitely much more to explore in vitro before performing experiments on mice. The last sentence (lines 113-114) is definitely too far-fetched and (if true) would fit better at the end of the paper, not below the first figure. But I don’t see why the authors claim these results confirm MTND therapy will not lead to drug resistance.

Section 2.2 – I don’t see how did the authors get to these experiments from previous figure. Also, you cannot just start a paragraph like that without any introduction what was done, and on how many samples.

Section 2.3

Again, the section starts with a conclusion and it shouldn’t be this way.

Line 193 – how do the authors know the action of those drugs was synergistic? Was some method like Chou-Talalay used to establish drug synergy?

And again the authors show some random results without saying which cell line(s) it is and how was it treated.

Materials and methods

3.3 Immunofluorescence staining – please add details about the antibodies that were used including catalog numbers

3.6 In vivo experiments – the authors included only 3 animals in each group. I honestly don’t see a point in such study as these groups are too small, it is not possible to draw any conclusions from such a study and the results are not going to be statistically significant. For me such a study is unethical.

3.10 – Information of clinical cell lines – please arrange this section differently as at the moment, with all the bullet points, it looks like each bullet point is a different cell line, which I don’t think is correct.

The methods section is a bit messy – at first the authors speak about cell culture and treatments, then cell assays, then animal studies after which they describe another cell assay and then go back to animal studies.

The level of English is ok, only sometimes I couldn't understand what did the authors want to say. However, the authors should definitely be more objective and do not use too big words to describe their results. 

Author Response

Reviewer2

Comments and Suggestions for Authors

Overall comments

It goes without saying that glioma is one of the diseases that urgently need better treatment. The authors of this paper describe a drug combination they used to induce differentiation in glioma and compare it to commonly used drug – Temozolomide. Unfortunately, the paper is very poorly organized, it’s very hard to understand what was done, why and on what samples. First of all, we don’t know how did the authors select drugs for this combinatorial treatment and how were the drug concentrations established. The first figure shows some in vitro and some in vivo results, both are insufficient to draw any conclusions. Then the authors jump to RNA sequencing results, however they don’t say what cells were analyzed there (from patients or cultured or from mice?). Later on it is still very confusing, unfortunately I was too exhausted to try making sense out of it.

Abstract

  1. The abstract is not following the standard plan: introduction, methods, results, conclusions. It’s hard to say what are the results of this study and what was published before. Authors start writing about the results in the 3rd sentence. Please, re-write it in a clearer way. Also, be more concise as at the moment the abstract is over 250 words, which is too long.

The author’s answer: Thanks for your suggestion. We have made extensive modifications to the abstract.

Introduction

  1. 43-46 – please re-write this sentence to make it grammatically and logically correct

The author’s answer: We have revised the text to address your concerns and hope that it is now clearer. Please see page 1, lines 38–44 of the revised manuscript.

  1. The authors forgot to mention that another isoform of retinoic acid, 13-cis RA, is used to treat neuroblastoma (which is a pediatric solid tumor) in patients with minimal residual disease.

The author’s answer: We sincerely appreciate the valuable comments. We have made some changes to the introduction of our manuscript. Please see page 2, lines 66–68 of the revised manuscript.

Results and discussion

  1. Section 2.1 - design and mechanism of MTND should be moved to Introduction, it is not a result. Also, please explain whether this MTND is something the authors established or is it widely used. How were the drugs selected and the concentrations optimized? The mechanism is only described in Figure 1b, but should be more detailed also in the text of the manuscript as this is the reason for all experiments.

The author’s answer: Thanks for the reviewer's comments, which we have seriously considered. The introduction part is generally used to introduce the research background, the research status, the main problems and put forward the solutions to the problems. However, design and mechanism of MTND shows the result of our selection of drugs based on differentiation therapy. Therefore, we prefer to put it in the results section. Moreover, we have made modification to the introduction (page 2, lines 67–69). We hope our explanation will meet with your approval. Through literature research, we selected the following 11 small molecules as alternatives, which include Retinoic acid, Forskolin, CHIR99021, Y27632, ISX9, DAPT, PD0325901, Dorsomorphin, LDN193189, Purmorphamine, P7C3-A20. These molecules are mainly involved in cell proliferation, cell migration, neural differentiation, and neuroprotection. Classified all small molecules as described above. Then one or two small molecules from each of the four categories were selected to form different drug combinations. The effects of each drug combination were verified in-vitro cell experiments. We selected the final drug combination (Forskolin, Dorsomorphin, Purmorphamine, CHIR99021, P7C3-A20) according to the neural induction efficiency of the drugs. As shown in the figure below (Fig.1), the induction effect of MTND (CHIR99021, Dorsomorphin, P7C3-A20, Forskolin, Purmorphamine) therapy was significantly superior to other combinations, so we chose MTND as our final choice. In addition, MTND contains 10μM Forskolin, 1μM Dorsomorphin, 1μM Purmorphamine, 3μM CHIR99021 and 3μM P7C3-A20. These concentrations were determined through an extensive literature research. At present, our work is mainly to verify the effect of the above drugs on tumor treatment. In the following work, we will further optimize the drug concentration to improve the therapeutic efficiency. In the design of and mechanism of MTND section, the mechanism was described. Meanwhile, we showed the multi-targets therapeutic genotype alterations induced by MTND in section 2.3, where we have further explained the mechanism of action of MTND.

Figure 1. The morphology of tumor cells in different states. (bright field image). (a) The cell morphology of tumor cells in its normal state (without any drug treatment); (b) Tumor cell morphology under other drug combinations; (c) Tumor cell morphology under MTND treatment. Scale bar: 200μm.

  1. Figure 1 – a- this figure suggests cells are extracted from patient’s tumor, treated ex vivo and then transferred back to the patient. Based on the text I don’t think that’s the case, so please change it to be less confusing. Also, from this figure it looks like MTND is blocking cell proliferation which leads to cell differentiation. But lack of proliferation does not equal differentiation induction, so please be more specific.

The author’s answer: We have modified this figure according to the Reviewer’s suggestion.

  1. Section 2.2 – it’s crucial to explain what drug concentrations were used and why (how were they established, especially for drug combinations). Were these primary cell lines, or were they established earlier? You cannot say “more than half of the patients” if you only tested 3 patients (better say 2 out of 3). Where did this number 350% come from? I don’t see it anywhere on the graphs and authors are referring to Figure 2c, which is a would healing assay that is not including any numbers. Both text and figure is very confusing, the authors should read it carefully and correct it. What is a SHAM control?

The author’s answer: MTND contains 10μM Forskolin, 1μM Dorsomorphin, 1μM Purmorphamine, 3μM CHIR99021 and 3μM P7C3-A20. In question 4, we have explained the selection of drug combinations and drug concentrations. Human clinical glioma tumors were resected through surgery and collected for use. Tumors and surrounding cells were processed and separated by mechanical trituration, trypsinization, filtration, and centrifugation in order to obtain the primary tumor cells. The number of tumor cell passages used in our experiments were passage 2-5. The number 350% is the improved inhibition rate of MTND compared to control group. As suggested by the reviewer, we have corrected the “more than half of the patients” into “2 out of 3”. We feel sorry for our carelessness. In our resubmitted manuscript, the mistake have been revised. Thanks for your correction. In vivo experiments, mice were divided into three groups, containing MTND, TMZ and physiological saline solution. The physiological saline solution group was defined as SHAM control in our manuscript. Please see page 7, lines 231–232 of the revised manuscript.

.

  1. Line 110 – what do the authors mean by morphology studies? Were the mice euthanized and then tumor burden was analyzed on histological sections?

The author’s answer: We sincerely thank the reviewer for careful reading. We have corrected the “morphology studies” into “histological studies”, and the histological study in this manuscript refers to the HE staining of mouse brain slices. After treatments, mice were anesthetized with chloral hydrate and perfused with normal saline and following 4% PFA. At the end of perfusion, brain tissues were removed by craniotomy and then tumor burden was analyzed on histological sections.

  1. Figure 2 – a-which cell line is it? The labels in the text do not match those in the figure (b mixed with c, etc). g – how many mice are represented by each bar?

The author’s answer: Figure 2(a,b) showed the proliferation and migration of cell line T-36. We feel sorry for our carelessness. In our resubmitted manuscript, the labels in the text have been modified to corresponds to the figures. In figure 2g, each bar represented three mice.

  1. To me, it’s not understandable why the authors jumped from 2 in vitro experiments to an animal study. There is definitely much more to explore in vitro before performing experiments on mice. The last sentence (lines 113-114) is definitely too far-fetched and (if true) would fit better at the end of the paper, not below the first figure. But I don’t see why the authors claim these results confirm MTND therapy will not lead to drug resistance. Section 2.2 – I don’t see how did the authors get to these experiments from previous figure. Also, you cannot just start a paragraph like that without any introduction what was done, and on how many samples. Section 2.3-Again, the section starts with a conclusion and it shouldn’t be this way.

The author’s answer: In Results and Discussion, we arranged figures and content according to the research purposes. For example, section 2.2 mainly explores the effect of the drug on the proliferation and migration of tumor cells, section 2.3 is about the effect of the drug on cell cycle arrest. In section 2.2, we think that both in vitro cell proliferation experiment and in vivo tumor histology study are intended to illustrate the effect of the drug on the proliferation of tumor cells, so we tend to put them in the same section. This inappropriate expression (drug resistance) has been deleted. We have re-written this part according to the Reviewer’s suggestion. Please see page 3, lines 103–106 of the revised manuscript. We have also re-written this part according to the Reviewer’s suggestion. Please see page 4, lines 137–138 of the revised manuscript.

  1. Line 193 – how do the authors know the action of those drugs was synergistic? Was some method like Chou-Talalay used to establish drug synergy?

The author’s answer: We sincerely thank the reviewer for careful reading. We have corrected the “Synergistic action” into “Combined action”. We were really sorry for our careless mistakes. We do not claim that these drugs are synergistic. This drug combination was selected after testing and showed the best induction effect.

  1. And again the authors show some random results without saying which cell line(s) it is and how was it treated.

The author’s answer: Specific details have been added to the manuscript. Please see page 4, lines 137–138 of the revised manuscript.

Materials and methods

  1. 3 Immunofluorescence staining – please add details about the antibodies that were used including catalog numbers.

The author’s answer: As suggested by the reviewer, we have added more details about the antibodies in M&M. Please see page 10, lines 323–326 of the revised manuscript.

  1. 6 In vivo experiments – the authors included only 3 animals in each group. I honestly don’t see a point in such study as these groups are too small, it is not possible to draw any conclusions from such a study and the results are not going to be statistically significant. For me such a study is unethical.

The author’s answer: Thanks to the professional advice of the reviewers, we have searched some relevant studies, all of which were conducted with three samples. Therefore, we believe that in the preliminary study, three repetitions of the experiment can still be recognized by most researchers, but later we will increase the number of animals for further verification.

  1. 10 – Information of clinical cell lines – please arrange this section differently as at the moment, with all the bullet points, it looks like each bullet point is a different cell line, which I don’t think is correct.

The author’s answer: As suggested by the reviewer, we have been rearranged this section. Please see page 11–12, lines 396–426 of the revised manuscript.

  1. The methods section is a bit messy – at first the authors speak about cell culture and treatments, then cell assays, then animal studies after which they describe another cell assay and then go back to animal studies.

The author’s answer: We sincerely appreciate the valuable comments. We have re-written this part according to the Reviewer’s suggestion.

Comments on the Quality of English Language

The level of English is ok, only sometimes I couldn't understand what did the authors want to say. However, the authors should definitely be more objective and do not use too big words to describe their results.

Reviewer 3 Report

In this paper, entitled “Muti-target neural differentiation (MTND) therapeutic cocktail to suppress brain tumor” by Xiaoping Hu et al., it is suggested that (MTND) therapeutic cocktail reduce glioma proliferation and could achieve effective treatment of brain tumor through unblocking their endogenous differentiation. Differentiation therapy, in cancer, aims to force the more undifferentiated cells of high malignancy to resume the process of maturation into differentiated cells of low tumorigenic potential. The authors also suggest that this method could provide a new treatment modality in glioma. Overall the concept of targeting tumor stem cells with therapies aiming at differentiating cells it is very interesting and has shown promising results in other type of cancer including brain cancer such as medulloblastoma and also in experimental models. In order to bring new knowledge to the field, suggested treatment modality should also be tested in MB organoids. Or at least in spheroids originating from glioma patients not previously cultured adherently in FBS, as these models better mimic the clinical situation and to avoid in vitro artefacts caused by long-term culturing in FBS and adherent growth.

MAJOR

GENERAL CONCEPT

The therapeutic efficacy of MTND treatment should be further analyzed, specifically regarding the cell cycle exit and entry into G0. It is important to investigate whether or not the cells in G0 are cancer stem cells. When a cell exits the cell cycle and enters the quiescent state known as the G0 phase, it may possess stem cell characteristics. During G0, stem cells temporarily halt their cell division and become inactive. However, upon treatment release and exposure to environmental challenges, these quiescent cells can resume their tumorigenic activity, potentially contributing to tumor relapse due to their tumor-initiating capabilities.

Differentiation therapy has been shown to reduce the proliferation potential of cancer cells, albeit in the short-term experimental setting, without inducing cell death. Therefore, it is important for the authors to investigate whether differentiation therapy leads to an improvement in the effectiveness of standard-of-care treatments, specifically by increasing apoptosis and thus reducing the presence of resistant cells.

The fact that the authors show improvement of brain functionality is impressive but it is not clear for me if that effect is directly due to a cancer cells that re-acquire a proper and specific brain cells functionality or if improve the functionality of the non-cancer cells present in the brain.

In the last figure panel a indicate that as for 72h also after 8 days of  in-vitro treatment there is reduction in cell proliferation that do not really means or prove that there is reduction of resistance. The author should water dares that with proper experiment or reconsider this sassumption providing more cornet interpretation of results.

Minor:

Fig.2

The scratch assay, is a commonly used in vitro technique to study cell migration and wound closure. While it is a relatively simple and cost-effective method, it does have several limitations in term of inaccuracy and variability, lack of physiological relevance, Cells in 2D cultures behave differently from cells in a native tissue environment, affecting their migration and healing responses. This result should be confirmed with additional approach such as transwell migration assay or live-cell imaging and tracking.

Figure 4

The quality of imagines should be improved and scale Barr included in the figure.

Figure 4c represent a functional category or a gene that fit into a specific functional category?

Figure5

which cell line is used forvexperiment

Facs quantification of annexinV/PI will be more precise and scan Barr should be integrated in the figure.

All figure legends should be carefully re-checked there are several editing mistake relative to the number of figure and what they represent. For example fig 2C is 2B.

If I can point an aspect that could be improved, I am not sure the relevant current literature on cancer cell differentiation in brain has been fully covered in the Introduction and Discussion sections.    

In general the paper show several editing mistake especially in figure and legends and professional English editing should be required.

Author Response

Comments and Suggestions for Authors

In this paper, entitled “Muti-target neural differentiation (MTND) therapeutic cocktail to suppress brain tumor” by Xiaoping Hu et al., it is suggested that (MTND) therapeutic cocktail reduce glioma proliferation and could achieve effective treatment of brain tumor through unblocking their endogenous differentiation. Differentiation therapy, in cancer, aims to force the more undifferentiated cells of high malignancy to resume the process of maturation into differentiated cells of low tumorigenic potential. The authors also suggest that this method could provide a new treatment modality in glioma. Overall the concept of targeting tumor stem cells with therapies aiming at differentiating cells it is very interesting and has shown promising results in other type of cancer including brain cancer such as medulloblastoma and also in experimental models. In order to bring new knowledge to the field, suggested treatment modality should also be tested in MB organoids. Or at least in spheroids originating from glioma patients not previously cultured adherently in FBS, as these models better mimic the clinical situation and to avoid in vitro artefacts caused by long-term culturing in FBS and adherent growth.

MAJOR

GENERAL CONCEPT

  1. The therapeutic efficacy of MTND treatment should be further analyzed, specifically regarding the cell cycle exit and entry into G0. It is important to investigate whether or not the cells in G0 are cancer stem cells. When a cell exits the cell cycle and enters the quiescent state known as the G0 phase, it may possess stem cell characteristics. During G0, stem cells temporarily halt their cell division and become inactive. However, upon treatment release and exposure to environmental challenges, these quiescent cells can resume their tumorigenic activity, potentially contributing to tumor relapse due to their tumor-initiating capabilities.

The author’s answer: As shown in Fig. 4b, the expression of Nestin of MTND-treated cells gradually disappeared. Studies showed that Nestin is a marker of cancer stem cells [1,2]. Therefore, we think the cells in G0 are not cancer stem cells.

[1] Neradil, J.et al., Nestin as a marker of cancer stem cells. Cancer Sci 106, 803–811 (2015).

[2] Wu, B., et al., Do relevant markers of cancer stem cells CD133 and Nestin indicate a poor prognosis in glioma patients? A systematic review and meta-analysis. J Exp Clin Cancer Res 34, 44 (2015).

  1. Differentiation therapy has been shown to reduce the proliferation potential of cancer cells, albeit in the short-term experimental setting, without inducing cell death. Therefore, it is important for the authors to investigate whether differentiation therapy leads to an improvement in the effectiveness of standard-of-care treatments, specifically by increasing apoptosis and thus reducing the presence of resistant cells.

The author’s answer: Thank you very much for pointing out this important issue. In section 2.3, we have verified the low cytotoxic pressure of MTND in the short-term experimental setting. Unfortunately, due to the limited time, we have not established clinical tumor-resistant cell lines for long-term monitoring and exploration the effect of MTND therapy in increasing apoptosis and reducing resistant cells. But later we will established clinical tumor-resistant cell lines for further verification.

  1. The fact that the authors show improvement of brain functionality is impressive but it is not clear for me if that effect is directly due to a cancer cells that re-acquire a proper and specific brain cells functionality or if improve the functionality of the non-cancer cells present in the brain.

The author’s answer: Thank you very much for this important question. Sorry for the lack of discussion of relevant content. In the brain, the main cells involved in memory and learning are nerve cells, which basically do not proliferate. The two main targets of MTND are induction of neural differentiation and cell cycle arrest, so we think its influence on the normal nerve cells in the brain should be less than the tumor cells. Therefore, we supposed that that effect is mainly due to a cancer cells that re-acquire a proper and specific brain cells functionality. Of course, the firm conclusions should be verified by further experimental design. In the future, we will increase the research exploration of relevant parts.

  1. In the last figure panel a indicate that as for 72h also after 8 days of in-vitro treatment there is reduction in cell proliferation that do not really means or prove that there is reduction of resistance. The author should water dares that with proper experiment or reconsider this sassumption providing more cornet interpretation of results.

The author’s answer: Thank you for your comments. We have re-written this part according to the Reviewer’s suggestion. Please see page 8, lines 377–483 of the revised manuscript. Here we do not mean reduction of resistance. By comparing the two reagents, we can see that MTND still has an inhibitory effect on cell proliferation after 8 consecutive days of treatment, while TMZ has initially shown an upward trend in cell proliferation. Therefore, we deduced that MTND could maintain a higher therapeutic effect than TMZ after continuous treatment without inducing cell resistance.

Minor:

  1. 2 The scratch assay, is a commonly used in vitro technique to study cell migration and wound closure. While it is a relatively simple and cost-effective method, it does have several limitations in term of inaccuracy and variability, lack of physiological relevance, Cells in 2D cultures behave differently from cells in a native tissue environment, affecting their migration and healing responses. This result should be confirmed with additional approach such as transwell migration assay or live-cell imaging and tracking.

The author’s answer: The scratch assay does have shortcomings, but we can see that in our experiment, it can still illustrate the effect of drugs on cell migration. As suggested by the reviewer, if the experimental method cannot explain the problems we need to study in the future, we will further improve the migration experiment and methods to related data acquisition.

  1. Figure 4 The quality of imagines should be improved and scale Barr included in the figure.

The author’s answer: We sincerely appreciate the valuable comments. We have modified this figure according to the Reviewer’s suggestion.

  1. Figure 4c represent a functional category or a gene that fit into a specific functional category?

The author’s answer: This figure presents functional categories associated with neurogenesis, synaptic transmission and axon genesis were upregulated after MTND treatment.

  1. Figure5 which cell line is used for experiment

The author’s answer: T-26 was used for the drug resistance test and cell apoptosis/necrosis analysis in Figure 5.

  1. Figure5 Facs quantification of annexinV/PI will be more precise and scan Barr should be integrated in the figure. All figure legends should be carefully re-checked there are several editing mistake relative to the number of figure and what they represent. For example fig 2C is 2B.

The author’s answer: We feel sorry for our carelessness. In our resubmitted manuscript, the labels in the text have been modified to corresponds to the figures.

  1. If I can point an aspect that could be improved, I am not sure the relevant current literature on cancer cell differentiation in brain has been fully covered in the Introduction and Discussion sections.

The author’s answer: We sincerely appreciate the valuable comments. We have checked the literature carefully and added more references on cancer cell differentiation into the INTRODUCTION part in the revised manuscript. Please see page 2, lines 67–69 of the revised manuscript.

Reviewer 4 Report

Broad comments

The work is very interesting and the quality is good, starting from the introduction which discuss and the state of the art and the aims of the research very well.

The study addresses a problem of great interest to the scientific community for the production of drugs with a broad spectrum in the field of molecularly targeted drugs. 

Currently, the critical point of target therapy is represented by the pleiotropism of tumor cells, which can simultaneously express different downstream or upstream markers of a specific molecular pathway. For this reason, especially for brain tumors such as glioblastoma (GBM), where inter- and intra-tumor heterogeneity prevails, the present study may assume considerable importance.

The study shows that Muti-Target Neural Differentiation (MTND) therapy can open up interesting perspectives in solving these problems.

The authors emphasize the role of MTND in avoiding cell anaplasia, which is a major hallmark of cancer cells.

In conclusion, the authors demonstrate that MTND is associated with improvements in learning and memory functions attributable to reprogramming of neuronal differentiation. These results may therefore be promising also in consideration of a therapy that avoids side effects.

 Specific comments:

1. I have some doubts about figure 1 a and b, where the mechanism of action of MTND is schematically represented. The authors explain that some substances can act on the neuronal cell, correctly reprogramming the cell cycle and directing them to cell differentiation. From the picture and the description it is not clear the role of the individual substances and their effect. Furthermore, purmophamine is not mentioned in the text but only in the image.

2. From the figure, it seems that CHIR99021 would block GSK-3β preventing neuronal differentiation. Similarly, Dorsomorphin would block SMAD and AMPK avoiding neogenesis and growth block. Therefore, I would propose that the authors briefly describe the individual role of the pathways and how blocking them would lead to the results.

3. The letters in figure 2 are not arranged correctly and this makes it difficult to interpret the data in the figure.

4. In lines 101 - 103 “However, more than half of patients showed an obvious resistance to TMZ and no significant inhibited cell proliferation was found in these TMZ 102 treated cells”. I don't understand which half of patients authors are referring to if a small number of biopsy samples templates are used.

5. Why was drug administration via brain stereotaxic tube selected? This administration does not take into account the pharmacodynamics of the drugs used, since they are inserted topically.

6. I suggest that the authors reduce the content of the description of the results in Figure 3 and 4. Some of these descriptions should be moved into the discussion.

7. I recommend improving the resolution of the microscope images.

8. I suggest writing a discussion section of the results that would help streamline the description of the figures.

9. The authors should clarify the use of the different cell lines:

a) Which cell line was used for the in vivo models?

b) Which cell lines were used in the various experiments? Only in Figure 2 is there an indication of the different cell lines.

c) The T-26 line in which experiment was it used?

10. The study opens various research perspectives; one suggestion would be to investigate metabolic changes in vitro and in vivo. The results concerning the upregulation of the neurogenesis, synaptic transmission and axon genesis pathways can be compared with studies evaluating the interaction between the Sonic hedgehog pathway and cellular communication processes in the GBM.(Torrisi F, et al. Biology (Basel). 2021 Aug 12;10(8):767. doi: 10.3390/biology10080767. PMID: 34439999)

I suggest to better explain the concept expressed in some sentences with short and simple words. Clarity and consistency of some sentences may be more helpful for readers

Author Response

Reviewer4

The work is very interesting and the quality is good, starting from the introduction which discuss and the state of the art and the aims of the research very well.

The study addresses a problem of great interest to the scientific community for the production of drugs with a broad spectrum in the field of molecularly targeted drugs. 

Currently, the critical point of target therapy is represented by the pleiotropism of tumor cells, which can simultaneously express different downstream or upstream markers of a specific molecular pathway. For this reason, especially for brain tumors such as glioblastoma (GBM), where inter- and intra-tumor heterogeneity prevails, the present study may assume considerable importance.

The study shows that Muti-Target Neural Differentiation (MTND) therapy can open up interesting perspectives in solving these problems.

The authors emphasize the role of MTND in avoiding cell anaplasia, which is a major hallmark of cancer cells.

In conclusion, the authors demonstrate that MTND is associated with improvements in learning and memory functions attributable to reprogramming of neuronal differentiation. These results may therefore be promising also in consideration of a therapy that avoids side effects.

Specific comments:

  1. I have some doubts about figure 1 a and b, where the mechanism of action of MTND is schematically represented. The authors explain that some substances can act on the neuronal cell, correctly reprogramming the cell cycle and directing them to cell differentiation. From the picture and the description it is not clear the role of the individual substances and their effect. Furthermore, purmorphamine is not mentioned in the text but only in the image.

The author’s answer: We sincerely appreciate the valuable comments. We have re-written this part according to the Reviewer’s suggestion. Please see page 2-3, lines 94–117 of the revised manuscript.

  1. From the figure, it seems that CHIR99021 would block GSK-3β preventing neuronal differentiation. Similarly, Dorsomorphin would block SMAD and AMPK avoiding neogenesis and growth block. Therefore, I would propose that the authors briefly describe the individual role of the pathways and how blocking them would lead to the results.

The author’s answer: Thank you for your comments. We have added some description in the introduction according to the Reviewer’s suggestion. Please see page 2-3, lines 94–117 of the revised manuscript.

  1. The letters in figure 2 are not arranged correctly and this makes it difficult to interpret the data in the figure.

The author’s answer: We feel sorry for our carelessness. In our resubmitted manuscript, the mistake have been revised.

  1. In lines 101 - 103 “However, more than half of patients showed an obvious resistance to TMZ and no significant inhibited cell proliferation was found in these TMZ treated cells”. I don't understand which half of patients authors are referring to if a small number of biopsy samples templates are used.

The author’s answer: Specific details have been added to the manuscript. Please see page 4, lines 126–129 of the revised manuscript.

  1. Why was drug administration via brain stereotaxic tube selected? This administration does not take into account the pharmacodynamics of the drugs used, since they are inserted topically.

The author’s answer: To ensure that our MTND therapy works more effectively, we treated mice with brain stereotaxic tube to achieve precise administration and improve treatment efficiency. We envision that the drugs could subsequently be encapsulated as a capsule for targeted administration. Of course, we will perform relevant experiments to study its pharmacokinetics in the future.

  1. I suggest that the authors reduce the content of the description of the results in Figure 3 and  Some of these descriptions should be moved into the discussion.

The author’s answer: We sincerely thank the reviewer for careful reading. Our manuscript was written in four parts: Introduction, Results and Discussion, Materials and Methods, and Conclusion. Therefore, we added the discussion to the description of the results in the second part, which contributes to a longer explanation of the figures. However, we believe that this is also in line with the requirements of this journal, such as the following articles are also in the same format: Int. J. Mol. Sci. 2022, 23(1), 567; https://doi.org/10.3390/ijms23010567. We hope to get your approval.

  1. I recommend improving the resolution of the microscope images.

The author’s answer: We sincerely appreciate the valuable comments. We have modified this figure according to the Reviewer’s suggestion.

  1. I suggest writing a discussion section of the results that would help streamline the description of the figures.

The author’s answer: We sincerely appreciate the reviewer’s comment. We have already answered this question in the sixth question and we hope our explanation will meet with your approval.

9 .The authors should clarify the use of the different cell lines:

  1. a) Which cell line was used for the in vivo models?
  2. b) Which cell lines were used in the various experiments? Only in Figure 2 is there an indication of the different cell lines.
  3. c) The T-26 line in which experiment was it used?

The author’s answer: We sincerely appreciate the valuable comments. We have added specific details in manuscript. a) T-36 were used for in vivo experiments. b) T-36, T-51, T-59 were used for cell proliferation assay and cell migration assay, while T-26 were used for RNA-seq, immunofluorescence staining experiment, drug resistance experiment, and cell apoptosis assay   in this manuscript. c) T-26 were used for RNA-seq, immunofluorescence staining experiment, drug resistance experiment, and cell apoptosis assays. Specific details above have been added to M&M.

  1. The study opens various research perspectives; one suggestion would be to investigate metabolic changes in vitro and in vivo. The results concerning the upregulation of the neurogenesis, synaptic transmission and axon genesis pathways can be compared with studies evaluating the interaction between the Sonic hedgehog pathway and cellular communication processes in the GBM.(Torrisi F, et al. Biology (Basel). 2021 Aug 12;10(8):767. doi: 10.3390/biology10080767. PMID: 34439999)

The author’s answer: Thank you for your suggestion, this is exactly what we need to work on in the future, and we are also working on this. You will see research in this area in our future work. We have carefully studied the literature you shared. The article explored the effect of sonic hedgehog regulation on the characteristics of glioblastoma induced by connexin 43 and revealed the interaction between connexin 43 and sonic hedgehog pathway in glioblastoma cell proliferation and migration. In our manuscript, we only considered the role of Shh signaling pathway in neurogenesis. Thank you very much for pointing out this important issue. We have added related content in our resubmitted manuscript. Please see page 3, lines 104–106 of the revised manuscript.

Round 2

Reviewer 1 Report

authors' responses are satisfactory

Reviewer 3 Report

The paper. improve during revision. The messages interesting. and fit well with the scope of the journal. Think should be accepted

minor editingisrequired